# Preparation and Performance of Silica/ESBR Nanocomposites Modified by Bio-Based Dibutyl Itaconate

**DOI:** 10.3390/polym11111820

**Published:** 2019-11-06

**Authors:** Haijun Ji, Hui Yang, Liwei Li, Xinxin Zhou, Lan Yin, Liqun Zhang, Runguo Wang

**Affiliations:** 1Beijing Advanced Innovation Center for Soft Matter Science and Engineering, State Key Laboratory of Organic-Inorganic Composites, Beijing Laboratory of Biomedical Materials, Beijing University of Chemical Technology, Beijing 100029, China; 2017210203@mail.buct.edu.cn (H.J.); 2017310124@mail.buct.edu.cn (H.Y.); 2018210268@mail.buct.edu.cn (L.L.); zhanglq@mail.buct.edu.cn (L.Z.); 2Jilin Petrochemical Research Institute, 27 Zunyi Road, Jilin 132000, China; jh_yinl@petrochina.com.cn

**Keywords:** nanocomposites, ESBR, dibutyl itaconate, hydrogen bonding interface, low rolling resistance

## Abstract

Ester-functionalized styrene-butadiene rubber (dibutyl itaconate-styrene-butadiene rubber) (D-ESBR) was synthesized by low-temperature emulsion polymerization using dibutyl itaconate (DBI) as a modified monomer containing ester groups. Nonpetroleum-based silica with hydroxy groups was used as a filler to enhance the D-ESBR, which can provide excellent mechanical properties, low rolling resistance, and high wet skid resistance. During the preparation of the silica/D-ESBR nanocomposites, a hydrogen-bonding interface was formed between the hydroxy groups on the surface of silica and the ester groups in the D-ESBR macromolecules. As the content of ester groups in the D-ESBR increases, the dispersion of silica in the nanocomposites is gradually improved, which was verified by rubber process analyzer (RPA) and scanning electron microscopy (SEM). Overall mechanical properties of the silica/D-ESBR modified with 5 wt % DBI were improved and became superior to that of the non-modified nanocomposite. Compared with the non-modified silica/D-ESBR, the DBI modified silica/D-ESBR exhibited a lower tan δ value at 60 °C and comparable tan δ value at 0 °C, indicating that the DBI modified silica/D-ESBR had lower rolling resistance without sacrificing wet skid resistance.

## 1. Introduction

With the international calls for energy conservation, environmental protection, and emission reduction, a growing voice to make motor vehicles more fuel efficient has become widespread all over the world, thus calling for a high-performance tire (i.e., a green tire) [1]. A green tire could achieve a balance of low rolling resistance, good wet skid resistance, and low external rolling noise. As one of the main components of a tire, elastomeric material is the only part that directly touches the ground. Therefore, the performance of elastomeric material plays a decisive role for low rolling resistance and good wet grip resistance of a tire [2]. Emulsion polymerized styrene-butadiene rubber (ESBR) is one of the most widely used synthetic rubbers. Due to its molecular structure, polymerization methods, and so on, conventional ESBR has a wide molecular weight distribution, poor filler dispersion, large heat generation, high rolling resistance, and does not meet the requirements for a green tire [3,4,5]. Solution polymerized styrene-butadiene rubber (SSBR) with excellent overall performance is used to manufacture green tires in the industry. However, due to a conditioned synthetic process, low production efficiency, and high production cost, SSBR still cannot completely replace ESBR in tire manufacturing on a large scale in a short time [6,7]. Driven by requirements of the EU Tyre Labelling Regulation and the competition threat of SSBR, it is imperative to optimize the overall performance of ESBR via physical and chemical modifications such as the functionalization of ESBR macromolecules. 

Recently, the preparation of silica/polymer nanocomposites has received attention all over the world. For example, silica has been incorporated into polymers such as silica gels [8] and poly(ethylene terephthalate) [9,10] to prepare silica/polymer nanocomposites, aiming to improve their overall performance. With the proposal of green tires, nonpetroleum-based silica has become one of the most widely used fillers in tire tread rubber [11] as silica can endow the tire tread rubber with excellent overall performance (excellent mechanical properties, low rolling resistance, and good wet skid resistance). However, the surface of silica contains a large number of hydroxyl groups [12], resulting in poor dispersion of silica in hydrophobic rubbers and poor compatibility with the rubber matrix [13]. There are two main ways to solve this problem: (i) hydrophobation of the silica surface using a coupling agent [14,15] and the subsequent chemical bonding between the rubber molecules and the coupling agent [16,17], and (ii) improvement of interfacial interactions between rubber and silica by introducing functional groups to the ESBR molecular chains [18].

At present, the functional modification of traditional synthetic rubber has developed rapidly through (1) epoxidation [19,20,21,22,23], (2) grafting [24,25], and (3) copolymerization [18,26,27,28]. Among them, copolymerization modification is an important means for imparting special functions to rubber in emulsion polymerization. The copolymerization modification of ESBR refers to copolymerizing a third functional monomer, such as glycidyl methacrylate (GMA) [28], hydroxypropyl methacrylate (HPMA) [29], acrylonitrile (ACN) [29], or methyl diethylaminoethyl acrylate [30], during the process of ESBR synthesis. The effects of various functional monomers in silica/ESBR compounds have been investigated, and results show that hydroxylated methacrylates and acrylates are suitable third co-monomers in ESBR for improving interfacial interaction in the compounds. Qiao et al. [27] showed demonstrated an excellent performance of silica/GMA-ESBR compounds without coupling agents. Hyunsung Mun et al. [31] manufactured silica/GMA-ESBR compounds using wet masterbatch (WMB) technology to reduce viscosity and improve processability. However, the functional monomers used in previous studies were all petroleum-based monomers. In order to reduce the dependence on non-renewable petrochemical resources in the rubber industry, we chose bio-based functional monomers derived from biomass resources [32,33,34] instead of petroleum-based functional monomers. Wang et al. [35,36,37] prepared a bio-based elastomer by emulsion copolymerization of itaconate with diene. Lei et al. [38] used various itaconates to copolymerize with isoprene to prepare bio-based elastomers with various side lengths. The results showed that dibutyl itaconate is the optimal bio-based monomer to prepare the bio-based elastomer, and that it has excellent overall performance. Based on the results shown by Lei et al., this work used dibutyl itaconate as a functional monomer to modify ESBR. Different to previously introduced petroleum-based functional monomers, dibutyl itaconate has two ester groups located on different sides, which means dibutyl itaconate can introduce higher polar group density than those petroleum-based monomers under the same conditions. Thus, we believe that dibutyl itaconate can be used to modify the molecular structure of ESBR and to improve the overall performance of ESBR.

In this study, an ester-functionalized styrene-butadiene rubber (dibutyl itaconate-styrene-butadiene rubber) (D-ESBR) was synthesized through low-temperature emulsion polymerization, and silica was used as a filler to prepare silica/D-ESBR nanocomposites. We studied the effects of ester content on properties of silica/D-ESBR nanocomposites, including static and dynamic mechanical properties, dispersion of silica, and wear resistance. The hydrogen bonding formed between the hydroxyl groups on the surface of silica and the ester groups of D-ESBR can improve the dispersion of silica and enhance the interfacial interaction, resulting in a significant reduction of rolling resistance and heat generation of the silica/D-ESBR nanocomposites. This study also provides an effective strategy for preparing high-performance elastomeric nanocomposites by forming hydrogen bonding between the filler and elastomer matrixes. The proposal schematic for hydrogen bonding formed in silica/D-ESBR nanocomposites is shown in Figure 1.

## 2. Experimental Section

### 2.1. Materials

Dibutyl itaconate (DBI, 96%) was purchased from Sigma-Aldrich. All other chemicals and solvents used in the polymerization, including styrene (97%), butadiene (94%), rosin soap, phosphoric acid, potassium hydroxide, ethylene diamine tetraacetic acid (EDTA), sodium dinaphthylmethanedisulfonate (TAMOL), ferric ethylene diamine tetraacetic acid salt (Fe-EDTA), sodium hydrosulfite (SHS), sodium hydroxymethanesulfinate (SFS), tert-dodecyl mercaptan (TDDM), p-menthane hydroperoxide (PMH), and ethanol were kindly provided by Jilin Petroleum Company and were used as received. The precipitated silica (Ultrasil VN3) with a BET specific surface area of 175 m^2^/g was purchased from Degussa Chemical. All of the rubber additives were industrial grade and commercially available.

### 2.2. Synthesis of Poly(dibutyl itaconate-co-styrene-co-butadiene) (D-ESBR)

D-ESBR was prepared by a redox-initiated emulsion polymerization reaction based on the ingredients displayed in Table 1. The polymerization reaction is shown in Figure 2. First, all of the chemicals listed in Table 1, except SHS and PMH, were added into the sealed reaction bottle, and the mixture in the bottle was pre-emulsified at 25 °C for 4 h. Later, the SHS solution was injected into the reaction bottle to eliminate residual oxygen in the pre-emulsion, followed by injecting the PMH (initiator) solution into the mixture to start the reaction. The polymerization was allowed to proceed for 8 h at 5 °C, followed by adding diethylhydroxylamine to terminate the polymerization and obtain the D-ESBR latex. The latex was then flocculated with ethanol to obtain wet D-ESBR elastomer, which was dried in a vacuum oven for 24 h at 60 °C.

### 2.3. Preparation of the Silica/D-ESBR Nanocomposites

The compounding formulation for silica/D-ESBR nanocomposites is shown in Table 2. First, silica with the coupling agent bis-(g-triethoxysilylpropyl)-tetrasulfide (Si69), which is a bifunctional silane with both hydrolysable and organofunctional ends, was kneaded with the D-ESBR in a Haake internal mixer for 8 min. Second, the mixture obtained was treated for 5 min at 150 °C to facilitate further reaction between D-ESBR and silica, then taken out and cooled down to room temperature. Third, the other additives were kneaded with the mixture to obtain silica/D-ESBR compound. Finally, the silica/D-ESBR compound was cured in a XLB-D 350 × 350 hot press (Huzhou Eastmachinery Corporation, Huzhou, China) under 15 MPa at 150 °C to prepare silica/D-ESBR nanocomposite.

### 2.4. Measurements and Characterization

#### 2.4.1. Fourier Transform Infrared Spectroscopy (FTIR) Analysis

A Tensor 27 Fourier transform infrared (FTIR) spectrophotometer (Bruker, Rheinstetten, Germany) equipped with a Smart Orbit diamond-attenuated total reflection (ATR) accessory was employed for the studies. An FTIR spectrum was acquired by scanning a sample 32 times in the wavenumber range of 400–4000 cm^−1^ at a resolution of 4 cm^−1^.

#### 2.4.2. Nuclear Magnetic Resonance (NMR) Spectroscopic Analysis

^1^H NMR spectroscopy of D-ESBR was performed on a Bruker AV400 spectrometer (Bruker, Rheinstetten, Germany). The solvent was CDCl_3_ with traces of tetramethylsilane as an internal reference.

#### 2.4.3. Gel Permeation Chromatography (GPC) Analysis

The average molecular weight of D-ESBR was measured by gel permeation chromatography (GPC) on a Waters Breeze instrument equipped with three water columns (Steerage HT3 HT5 HT6E) using tetrahydrofuran as the solvent (1.0 mL/min) and a Waters 2410 refractive index detector (Water, Milford, MA, USA). Polystyrene standards were used for calibration.

#### 2.4.4. Differential Scanning Calorimetry (DSC) Analysis

Differential scanning calorimetry (DSC) of D-ESBR was conducted with a Mettler-Toledo DSC instrument (Mettler-Toledo International Inc., Greifensee, Switzerland) under nitrogen. The sample was heated to 100 °C and kept isothermal for 3 min to remove the previous history. Then, it was cooled to −100 °C and reheated to 100 °C. The heating (cooling) rate was 10 °C/min.

#### 2.4.5. Thermogravimetric Analysis (TGA)

Thermogravimetric analysis (TGA) of D-ESBR was conducted by using a STARe system TGA/DSC1 thermogravimeter (Mettler-Toledo International Inc., Greifensee, Switzerland) equipped with a cooling water circulator. The TGA thermograms were obtained under flowing nitrogen (50 mL/min) at the scanning rate of 10 °C/min in the temperature range of 30–800 °C. All of the samples were accurately weighed, and the weight of each sample was ~10 mg.

#### 2.4.6. Calculation of Cross-Linking Density

The cross-link densities of the D-ESBR nanocomposites weighed (*m*_0_) were measured by swelling experiments with toluene for a total of 72 h. After reaching equilibrium, the samples were taken out, wiped with filter paper, and weighed (*m*_1_). The samples were dried in an oven at 60 °C for 48 h to remove all the solvent and weighed again (*m*_2_). The swelling ratio Q was calculated using Equation (1):(1)$$Q=\frac{\left(m_{1}-m_{0}\right)/\rho_{1}}{m_{0}/\rho_{2}}$$
where *ρ*_1_ and *ρ*_2_ are the densities of the solvent and polymer, respectively.

To calculate the cross-linking density (*v_e_*) of the samples, Flory–Rehner expression was used. Equation (2) is given as follows [39,40]:(2)$$v_{e}=-\frac{\ln\left(1-v_{r}\right)+v_{r}+\chi v_{r}{}^{2}}{V_{s}\left(v_{r}{}^{1/3}-\frac{1}{2}v_{r}\right)}$$
where *χ* is the Flory–Huggins polymer solvent interaction parameter (for the SBR/toluene system, *χ* = 0.446), *V_s_* is the molar volume of the solvent (*V_toluene_* = 106.35 cm^3^/mol), and *v_r_* is the volume fraction of the polymer at equilibrium swelling, which was calculated using Equation (3) [41]:(3)$$v_{r}={\left[1+\left(\frac{m_{1}-m_{2}}{m_{2}}\right)\left(\frac{\rho_{1}}{\rho_{2}}\right)\right]}^{-1}$$

#### 2.4.7. X-ray Diffraction (XRD)

X-ray diffraction (XRD) measurements (Rigaku, Tokyo, Japan) were determined in the range of 5–40° at a scan rate of 5° min^−1^ by using a Rigaku D/Max 2500VBZt/PC X-ray diffractometer with Cu-Kα radiation (40 kV, 200 mA).

#### 2.4.8. Scanning Electron Microscopy (SEM) Analysis

The surface morphology of the silica/D-ESBR nanocomposites were observed using a Hitachi S-4800 scanning electron microscope (Hitachi Co., Tokyo, Japan). The samples were prepared by fracturing the composites in liquid nitrogen. The fracture surfaces were sputter coated with gold before observation.

#### 2.4.9. Rubber Process Analyzer (RPA) 

Strain sweep experiments were performed on uncured compounds and nanocomposites with an Alpha RPA 2000 rheometer (Alpha Technologies Co., Ltd., Akron, OH, USA) at 60 °C and 1 Hz. 

#### 2.4.10. Dynamic Thermomechanical Analysis (DMTA)

The dynamic mechanical thermal properties were implemented on a 01 dB-Metravib VA 3000 dynamic mechanical thermal analyzer (Rheometric Scientific Co, Limonest, France) at 10 Hz under the strain amplitude of 0.3% in a tension mode. In this mode, we adjusted the pretension by controlling “displacement”, in which the value is generally between −100 and −50, and the displacement is adjusted at −70 during the test. This displacement is represented the pretension as mentioned. The temperature was scanned from −80 to 100 °C with a heating rate of 3 °C/min.

#### 2.4.11. Physical and Mechanical Tests

Tensile tests were conducted on dumbbell-shaped specimens at 25 °C according to ASTM D412 by using a SANS CMT 4104 electrical tensile instrument (Shenzhen SANS Test Machine Co., Ltd., Shenzhen, China) at a crosshead speed of 500 mm/min. The Shore A hardness test was performed according to ASTM D2240 with a Bareiss HPE II hardness apparatus (Bareiss Prufgeratebau GmbH, Oberdischingen, Germany).

#### 2.4.12. Thermo-Oxidative Accelerated Aging Test

The dumbbell-shaped D-ESBR nanocomposite specimens were hung in the air-circulating cabinet oven GT-7017-E (Gotech Testing Machines Co., Ltd., Qingdao, China) at 100 °C for 24 and 72 h. The mechanical properties (tensile strength σ and elongation at break ε) were tested before and after aging. The aging coefficient *K* was used to evaluate thermo-oxidative aging properties [42]. Equation (4) is given as follows:(4)$$K=\frac{F}{F_{0}}$$
where *F* = σ × ε, and *F*_0_ and *F* represent the tensile product before and after aging, respectively.

## 3. Results and Discussion

### 3.1. Structure and Characterization of D-ESBR

D-ESBR with high molecular weight was successfully synthesized via a mild redox emulsion polymerization. The yield of D-ESBR was intentionally limited to 70%–80% in order to obtain optimum mechanical properties, since the D-ESBR with higher a yield than 80% has a high gel content. Actually, gels can be clearly observed when the yield reaches up to 80%. The average molecular weight (M_n_) and the polydispersity index (PDI) affect the mechanical and processing properties of rubber. Table 3 shows that the Mn of D-ESBR (5) and D-ESBR (10) is about 190,000 g/mol with a PDI of about 2.5, while the Mn of D-ESBR (0) is slightly larger than the above ones with a PDI of about 3.1. 

The chemical structure of D-ESBR was confirmed by FTIR and ^1^H NMR. Figure 3 shows the FTIR spectra of D-ESBR with various DBI contents. All spectra show the following similar absorptions: absorption peaks at 3023, 2917, and 2846 cm^−1^ are attributed to the stretching vibrations of –CH_3_, –CH_2_, and –CH groups in the copolymer, respectively; the absorptions between 1440 and 1490 cm^−1^ belong to the skeletal vibration of the benzene ring of the styrene unit; the absorption at 1639 cm^−1^ belongs to the C=C stretching vibration of the butadiene unit; the absorptions at 702 and 963 cm^−1^ correspond to the C–H deformation vibrations of the *cis*-1,4 units and *trans*-1,4 units of butadiene; and the absorptions at 906 cm^−1^ are attributed to the C–H deformation vibration of the vinyl structure of the butadiene unit. The difference in the spectra is the absorption at 1728 cm^−1^, which corresponds to the C=O stretching vibration of the DBI unit, and as the DBI content increases, the intensity of the absorption increases, indicating that DBI units were successfully introduced into the copolymers.

Macromolecular structure of D-ESBR was further confirmed by ^1^H NMR. The spectra and the detailed assignments of each peak to the molecular structure are shown in Figure 4. The small peaks at 4.05 ppm originate from the protons adjacent to the ester group. The results of the FTIR and ^1^H NMR verify that DBI was been successfully introduced into the D-ESBR chains.

The ester groups in D-ESBR are significant to the dispersion of silica because they are expected to form hydrogen bonds with the silanol on the surface of silica in the subsequent experiments. The actual DBI content of D-ESBR was estimated by the integral ^1^H NMR spectrum and is shown in Table 4. The results show that the DBI content in D-ESBR chains was consistently less than the initial DBI content fed into the polymerization system.

### 3.2. Thermal Properties of D-ESBR

The glass transition temperature (T_g_) of D-ESBR was determined using Differential scanning calorimetry (DSC) thermograms (Figure 5). The introduction of DBI had little effect on the T_g_ of the D-ESBR. Moreover, neither of the three curves show any crystallization peak, indicating that the D-ESBR were still amorphous.

TGA measurement of D-ESBR was carried out and is shown in Figure 6. For the TGA curves, the weight loss (6.2%) of D-ESBR (0) in the range of 200–330 °C was due to the decomposition of residual small molecules and oligomers, while the weight of D-ESBR (5) and D-ESBR (10) in that temperature range remained constant due to the suppression of oligomers by introducing DBI into the polymerization system. The decomposition temperature slightly decreased with an increased DBI content, suggesting that the introduction of DBI into the ESBR macromolecular chain weakened the thermal stability of D-ESBR. The main reason for the weakness of the thermal stability was the reduction of styrene content in the polymer composition due to the introduction of DBI.

### 3.3. XRD Patterns of the Silica/D-ESBR Nanocomposites

For the nonlayered silica nanoparticle composites, XRD was commonly performed to analyze the degree of crystallinity of the nanocomposites. Figure 7 shows the XRD patterns of the silica/D-ESBR nanocomposites with various DBI contents. The broad diffraction peak at around 21° indicates that the D-ESBR is amorphous without a crystalline structure, while the sharp peaks at 32° and 36° in the patterns relate to the additives such as zinc oxide. 

### 3.4. Curing Characteristics and Cross-Linking Densities of Silica/D-ESBR Nanocomposites

Figure 8 shows the characteristic curing curves of D-ESBR with various DBI contents at 150 °C. The curing time of silica/D-ESBR (5) and silica/D-ESBR (10) was longer than that of silica/D-ESBR (0) because of the probable effect between the ester groups and the accelerators. The torque of silica/D-ESBR (5) and silica/D-ESBR (10) was significantly lower than that of silica/D-ESBR (0) because filler–filler networks occur less often in silica/D-ESBR (5) and silica/D-ESBR (10). The torque difference (M_H_-M_L_) of the vulcanization curve was closely related to the cross-linking density of the vulcanizates. With the introduction of DBI, the torque difference gradually decreased, indicating a decrease of the cross-linking density of silica/D-ESBR.

To further investigate the effect of the ester groups and verify the above explanation, a calculation of swelling ratios and cross-linking densities was carried out using Equations (1) and (2), respectively, with the results are shown in Table 5. The structure of D-ESBR was mainly composed of styrene and butadiene, which is similar to that of SBR, so that the interaction parameters χ of SBR from polymer handbooks can be still used. As DBI content of D-ESBR increased, cross-linking density decreased and the swelling ratio increased. These results are attributable to the following factors: (1) cross-linking densities increase with improved interfacial interaction; (2) with the introduction of DBI, ester groups of DBI may absorb a portion of the accelerators, resulting in the formation of more polysulfide bonds and disulfide bonds in the curing system. An increase in total polysulfide bond density and disulfide bond density will lead to a decrease in cross-linking density, in which case the second factor dominates. 

### 3.5. Silica Dispersion and Interfacial Interaction of Silica/D-ESBR Nanocomposites

There are two kinds of filler networks in a filled rubber compound: one that is formed through filler–filler direct contact (filler–filler network), and one that is formed through macromolecular chain bridges anchored between neighboring fillers (filler–rubber network) [43,44]. Generally, the formation of filler–filler networks result in poor silica dispersion and interfacial interaction, while the formation of filler–rubber networks result in improved silica dispersion and interfacial interaction. A rubber process analyzer (RPA) allows the reliable investigation of filler–polymer interaction. The strain amplitude dependence of G’ in silica/D-ESBR compounds is shown in Figure 9a. Silica/D-ESBR (0) had a high initial G’ value and a pronounced Payne effect, indicating a very strong filler–filler network due to poor dispersion of silica in the nanocomposite. The introduction of DBI effectively reduced the initial G’ value and Payne effect, indicating that the filler–filler network was suppressed and the dispersion of silica was improved. As the DBI content increased, the initial G’ value of the silica/D-ESBR continued to decrease. The reason can be explained as follows: The introduction of DBI facilitated the formation of hydrogen bonds between the D-ESBR chain and the silica surface [35]. Hydrogen bonds can increase the interfacial interaction between D-ESBR and silica and suppress the filler–filler interaction, resulting in good dispersion of the filler. The slope of G’ for silica/D-ESBR (5) at low strain was significantly different than that of the other compounds (due to the hydrogen bonding, the filler–filler network of the silica/D-ESBR (5) was weakened, so the Payne effect was significantly weakened at low strain). Compared to the silica/D-ESBR (5), however, the silica/D-ESBR (10) had a higher hydrogen bond density, which also contributed to the G’ value. The weak hydrogen bonds were easily destroyed under low strain shear, resulting in greater decrease in G’ value than that of the silica/D-ESBR (5). The strain amplitude dependence of tan δ in the silica/D-ESBR compounds is shown in Figure 9b. Poor dispersion of silica results in a strong filler–filler network that was easily destroyed as strain increases. Destruction of the strong filler–filler network increased filler–filler friction, and weak interfacial interaction increased filler–rubber friction, which resulted in high tan δ at high strain. The introduction of DBI helped to weaken the filler–filler network and improved the interfacial interaction. Therefore, the tan δ of silica/D-ESBR (5) and silica/D-ESBR (10) was lower than that of silica/D-ESBR (0) at high strain. Among them, D-ESBR (10) had the lowest degree of cross-linking, so the value of tan δ is increased compared with ESBR (5), while also being obviously lower than D-ESBR (0). Generally, in the tire industry, the tan δ value at a strain of 7% of rubber composites is closely related to the tire rolling resistance [45]. For example, Qiao et al. reported that a rubber nanocomposite has a low tan δ value (0.120), indicating that the rubber nanocomposite could be used to manufacture a tire with a low rolling resistance [27]. For the silica/D-ESBR nanocomposites, the tan δ value at a strain of 7% are summarized in Table 6. It is worth noting that the tan δ value at a strain of 7% of silica/D-ESBR (5) was reduced to 0.102, indicating that the silica/D-ESBR nanocomposites are capable of manufacturing a tire with a low rolling resistance. In summary, DBI can improve the interfacial interaction between D-ESBR and silica, resulting in good dispersion of silica, which plays a positive role in reducing rolling resistance of a tire. The results show that D-ESBR elastomer has a great potential for green tire tread applications in future.

To further investigate the dispersion of silica in the silica/D-ESBR nanocomposites, representative fracture surfaces were studied by SEM. In Figure 10, the dark part represents the D-ESBR matrix and the light part represents the filler particles that have been densely distributed in the matrix. Figure 10 shows that the dispersion of silica in the silica/D-ESBR (5), and the silica/D-ESBR (10) is more homogenous than that of the silica/D-ESBR (0). As discussed above, the introduction of DBI containing ester groups is favorable to improving the interfacial interaction between D-ESBR and silica and the dispersion of silica in the silica/D-ESBR nanocomposites.

### 3.6. Mechanical Performance of Silica/D-ESBR Nanocomposites

Mechanical properties of silica/D-ESBR nanocomposites are displayed in Figure 11 and summarized in Table 7. The tensile strengths of D-ESBR with various DBI contents are excellent and similar, indicating that the introduction of DBI does not affect its tensile strength dramatically. In addition, D-ESBR (5) exhibits better overall mechanical properties than D-ESBR (0) because of its hydrogen bonding interaction and a good dispersion of silica (Figure 10). Compared with that of D-ESBR (0), the modulus at 300% strain of D-ESBR (5) increased by 16.7%, while the elongation at the break of D-ESBR (5) decreased due to a strong intermolecular interaction. The smallest permanent set in Table 7 indicates that D-ESBR (5) had the best recovery ability. The degeneration of the mechanical properties of D-ESBR (10) may have been attributed to a reduction in cross-linking density. 

Abrasion resistance and heat build-up are important indicators of rubber performance. The abrasion resistance and heat build-up of silica/D-ESBR nanocomposites was measured, and the results are displayed in Figure 12 and Table 7. With an increase in DBI content, the Akron wear gradually increased, while the compression heat generation gradually reduced.

### 3.7. Dynamic Mechanical Performance of Silica/D-ESBR Nanocomposites

The temperature dependence of the loss factor tan δ in silica/D-ESBR nanocomposites with different DBI contents is shown in Figure 13 and summarized in Table 8. As the DBI content increased, the highest tan δ value at T_g_ increased as well. The highest tan δ value at T_g_ for filled rubber nanocomposites was mainly affected by two factors: (1) the intensity of interfacial interaction, and (2) the filler dispersion status. For silica/D-ESBR nanocomposites, the two factors compete with each other. First, the presence of hydrogen bonds can enhance the interfacial interaction, resulting in a decreased highest tan δ value at T_g_. Second, the silica dispersion in the nanocomposites is improved because of the introduction of ester groups, resulting in a weak filler–filler network, facile chain segmental relaxation, and an increased highest tan δ at T_g_. The latter dominates in silica/D-ESBR nanocomposites. It is widely accepted in the tire industry that a high tan δ at 0 °C indicates good wet skid resistance, while a low tan δ at 60 °C indicates a low rolling resistance [46]. Compared to silica/D-ESBR (0), silica/D-ESBR (5) had a lower tan δ at 60 °C, indicating lower rolling resistance. The tan δ values at 60 °C of the silica/G-ESBR nanocomposites was between 0.108 and 0.123 [31]. However, D-ESBR (10) had comparable tan δ values to silica/D-ESBR (0) at 0 and 60 °C because of the low cross-linking density (Table 5), indicating that the wet skid resistance and rolling resistance of silica/D-ESBR (10) were not significantly improved.

### 3.8. Thermo-Oxidative Aging Performance of D-ESBR Nanocomposites

The effect of the thermo-oxidative aging parameter is very important in outdoor applications such as tire. Figure 14 presents the mechanical properties of D-ESBR nanocomposites before and after thermo-oxidative aging, and their aging coefficients at 100 °C for 24 and 72 h are shown in Figure 15. The silica/D-ESBR (5) and silica/D-ESBR (10) nanocomposites showed a higher aging coefficient than silica/D-ESBR (0), indicating that the introduction of DBI can improve the anti-aging ability of silica/D-ESBR nanocomposites, since the introduction of DBI decreases the double bond content in D-ESBR macromolecules.

## 4. Conclusions

Based on bio-based dibutyl itaconate-containing ester groups, poly(dibutyl itaconate-co-styrene-co-butadiene) (dibutyl itaconate-styrene-butadiene rubber) (D-ESBR) was successfully synthesized by redox emulsion polymerization, and the silica/D-ESBR nanocomposites were prepared. After the ester group was introduced into the D-ESBR, the dispersion of silica in the rubber matrix and the interfacial interaction between silica and rubber chains were improved. The improvements were attributed to hydrogen bonds between the hydroxyl groups of silica and the ester groups of D-ESBR, which play a positive role in the reduction of rolling resistance. In addition, the physical and mechanical properties of silica/D-ESBR nanocomposites with different DBI contents were studied. The results showed that D-ESBR with 5 wt % DBI had excellent overall mechanical properties, low heat generation, the lowest rolling resistance, comparable wear resistance and the best anti-aging performance among the tested nanocomposites. The deteriorated performance of D-ESBR with 10 wt% DBI was due to a reduction in cross-linking density. All in all, small quantities of DBI were used to introduce the ester groups into D-ESBR macromolecular chains, which can improve the overall performance of D-ESBR and provide a new strategy for “high performance” (green) automotive tire materials.

## Figures and Tables

**Figure 1 polymers-11-01820-f001:**
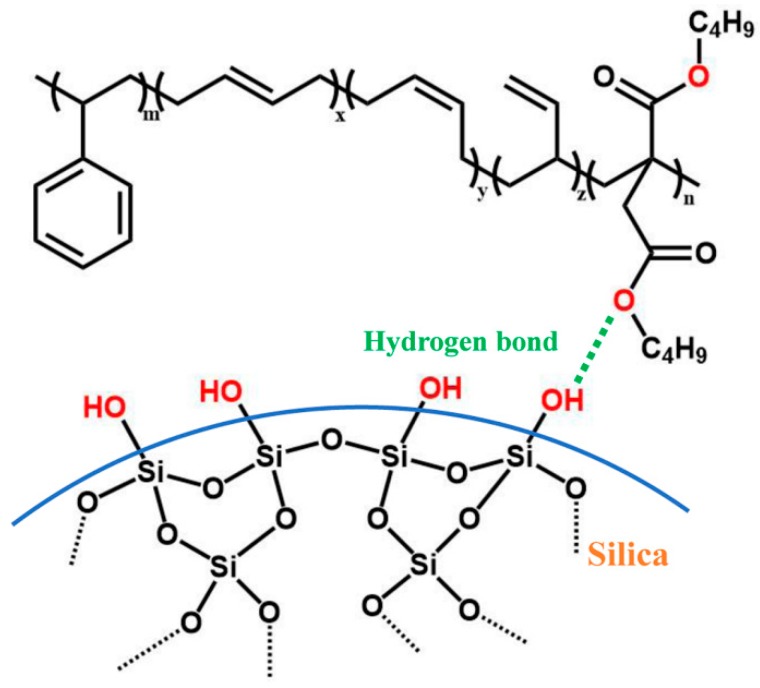
Proposal schematic of hydrogen bonding formed between silica and D-ESBR.

**Figure 2 polymers-11-01820-f002:**
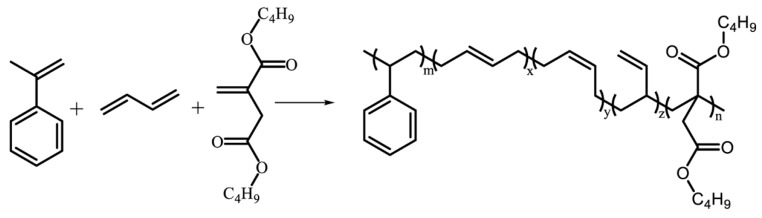
Polymerization reaction of D-ESBR with various DBI contents.

**Figure 3 polymers-11-01820-f003:**
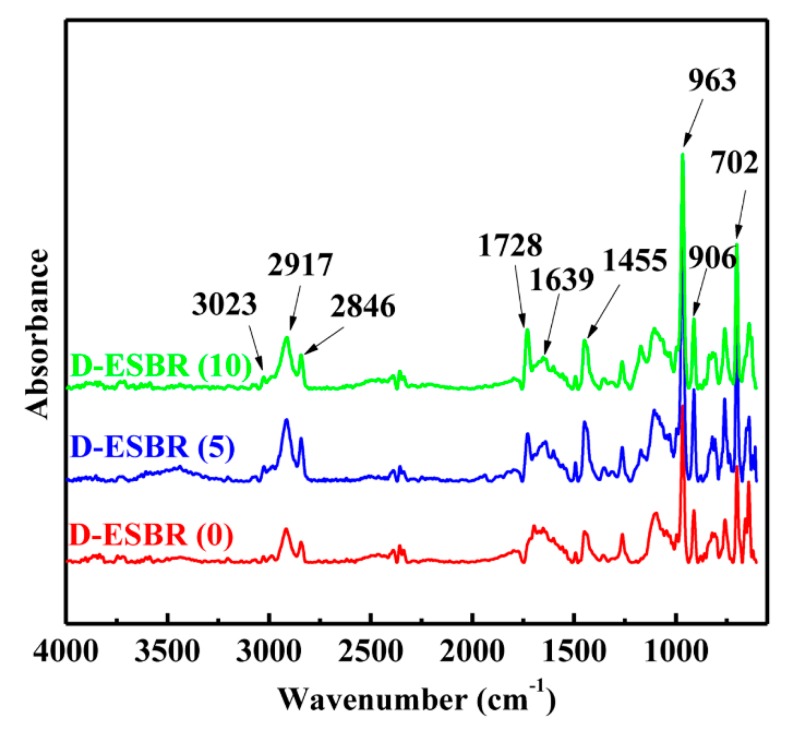
FTIR spectra of D-ESBR with various DBI contents.

**Figure 4 polymers-11-01820-f004:**
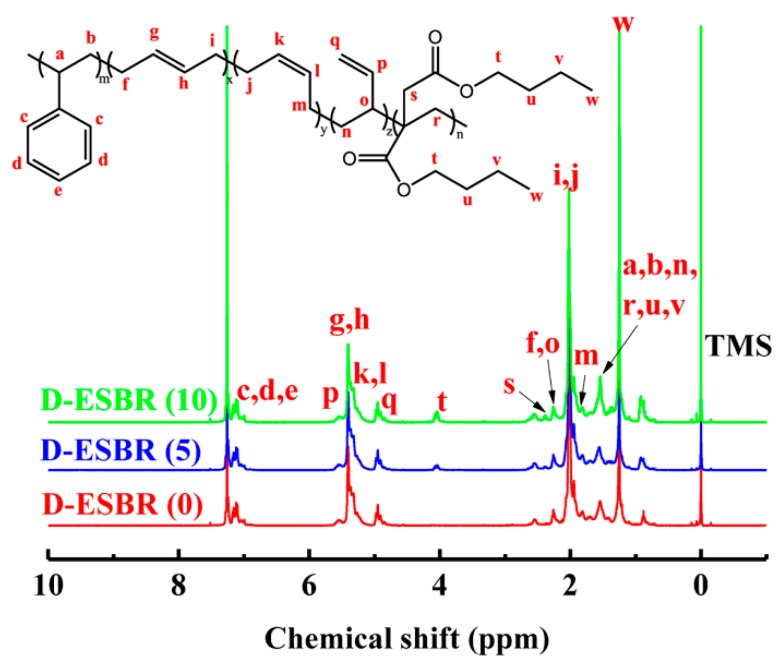
^1^H NMR spectra of D-ESBR with various DBI contents (in CDCl_3_).

**Figure 5 polymers-11-01820-f005:**
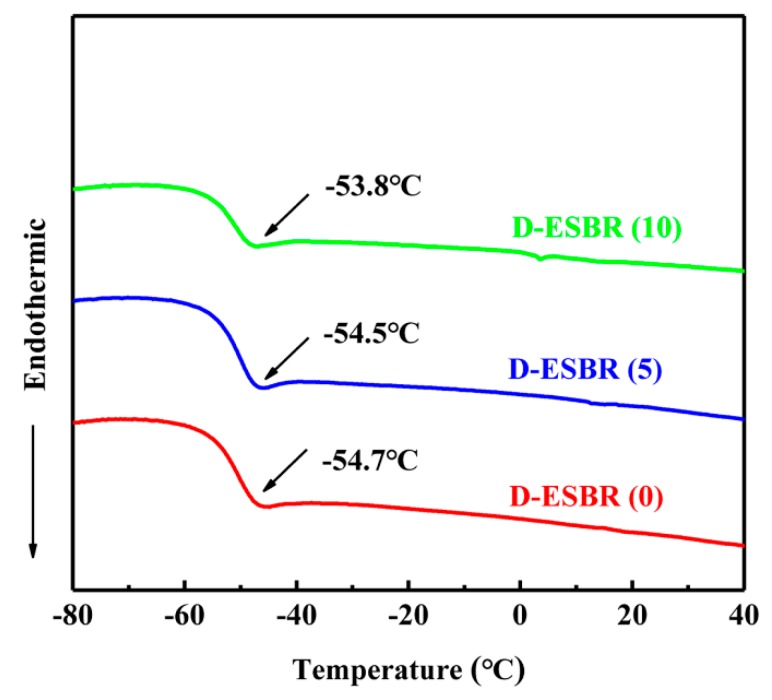
DSC thermograms of D-ESBR.

**Figure 6 polymers-11-01820-f006:**
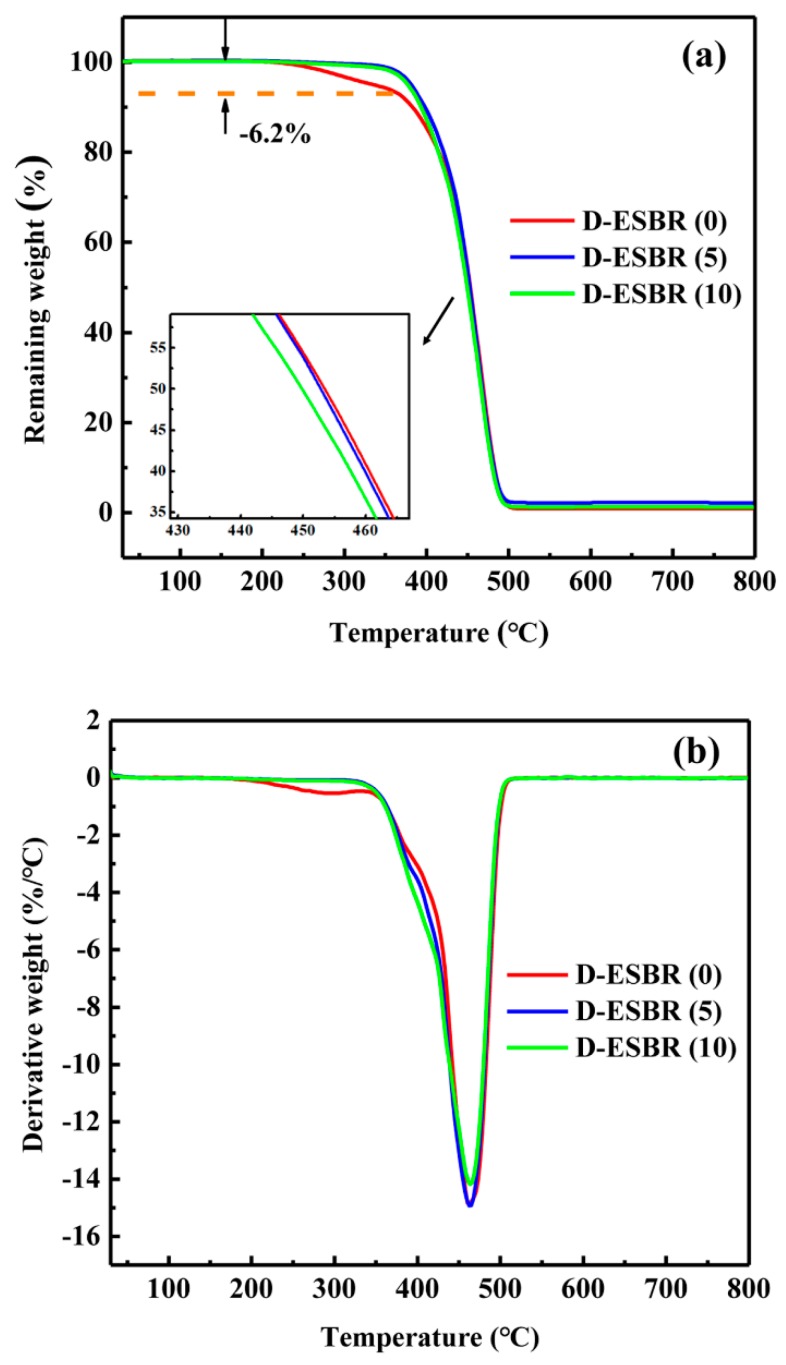
(**a**) TGA and (**b**) TGA derivative curves of D-ESBR.

**Figure 7 polymers-11-01820-f007:**
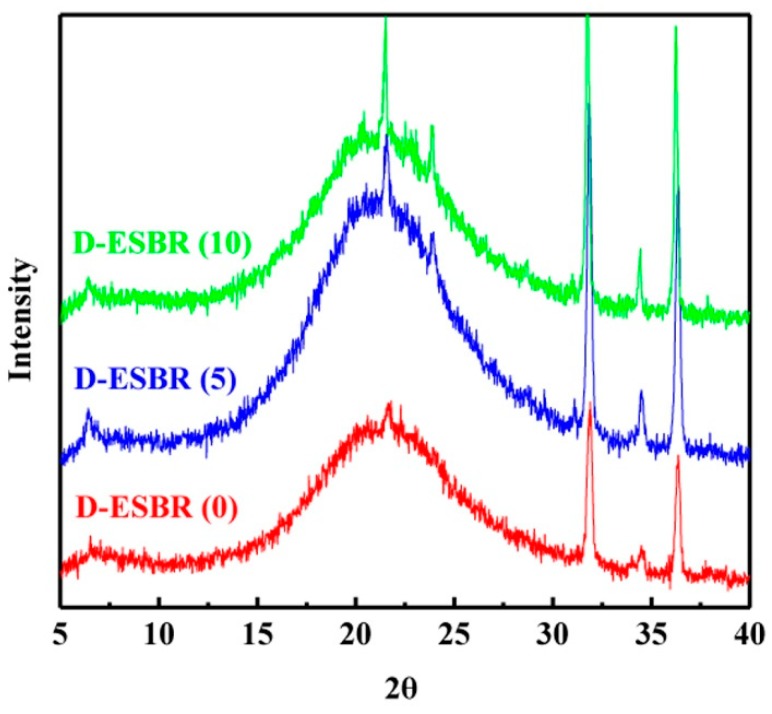
XRD patterns of the silica/D-ESBR nanocomposites with various DBI contents.

**Figure 8 polymers-11-01820-f008:**
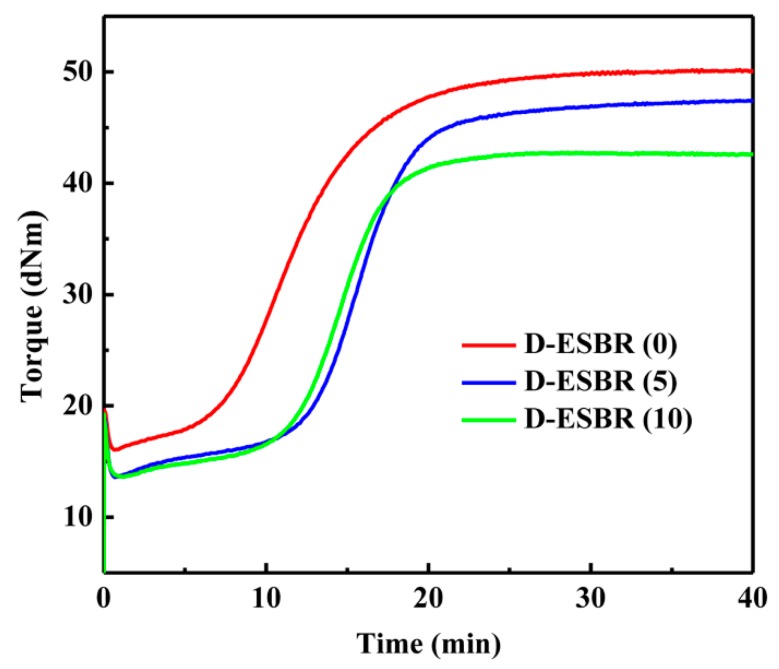
Curing curves for the silica/D-ESBR nanocomposites.

**Figure 9 polymers-11-01820-f009:**
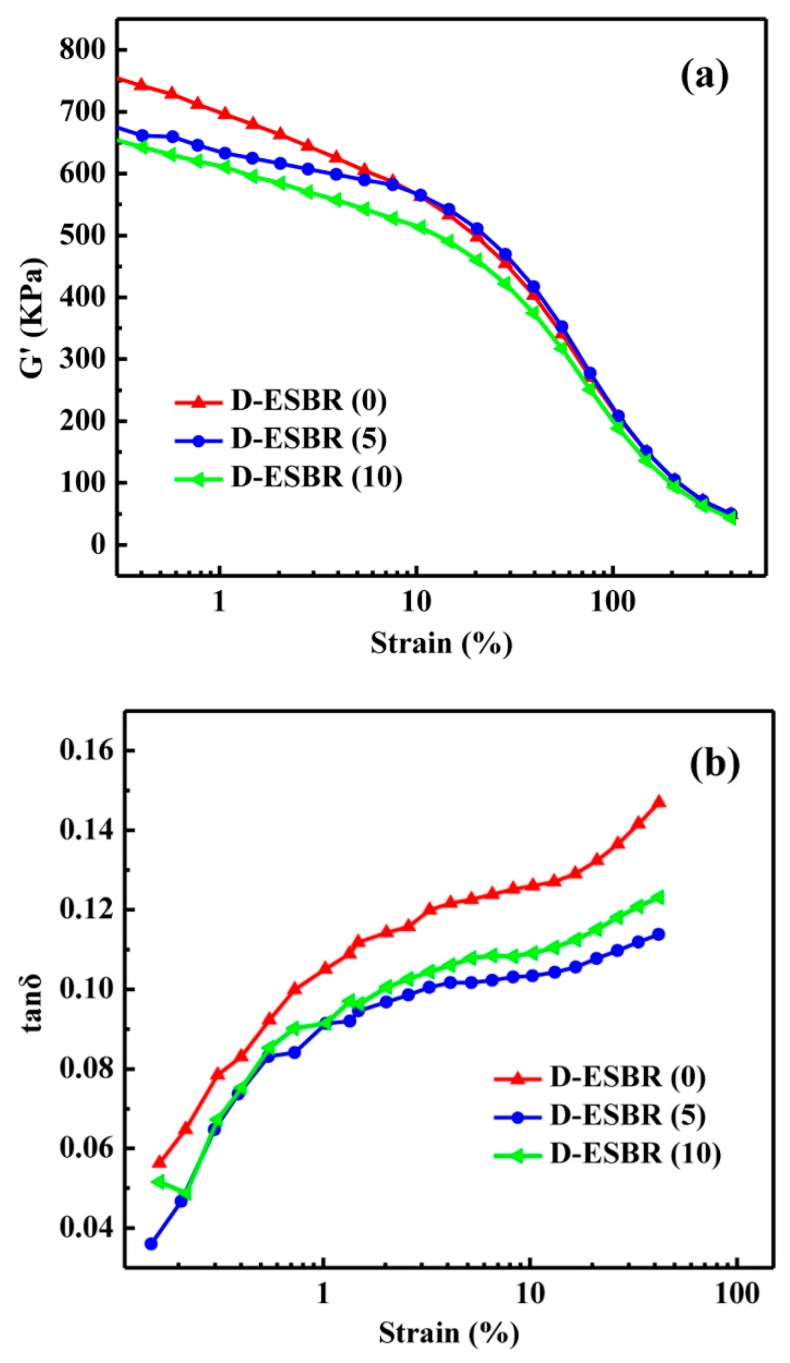
Strain amplitude dependence of (**a**) G′ of un-vulcanized D-ESBR with various DBI contents, and (**b**) tan δ of vulcanized D-ESBR with various DBI contents.

**Figure 10 polymers-11-01820-f010:**
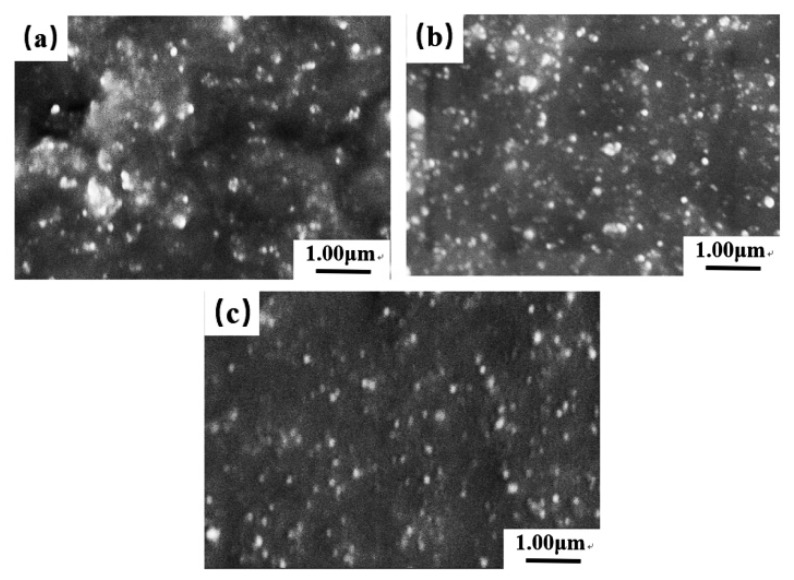
SEM micrographs of (**a**) silica/D-ESBR (0), (**b**) silica/D-ESBR (5), and (**c**) silica/D-ESBR (10) nanocomposites.

**Figure 11 polymers-11-01820-f011:**
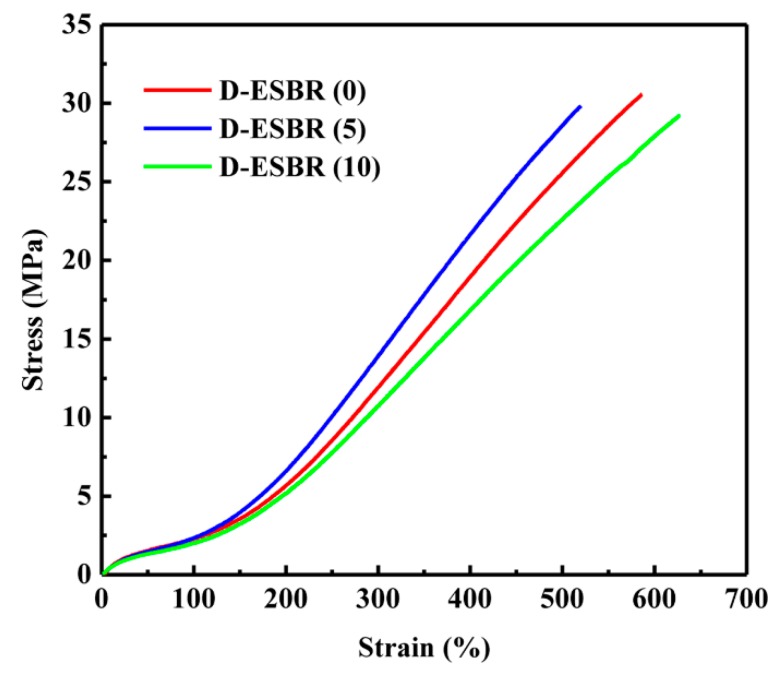
Stress-strain curves of the silica/D-ESBR nanocomposites.

**Figure 12 polymers-11-01820-f012:**
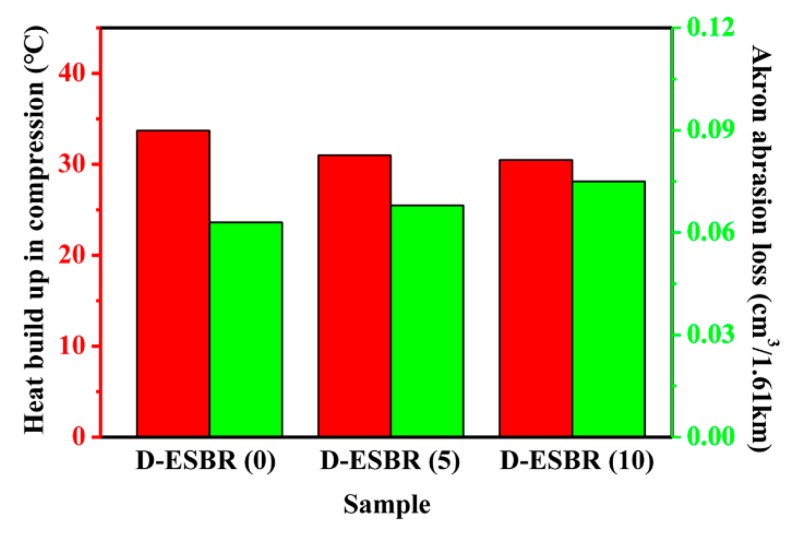
Akron abrasion loss and heat build-up of D-ESBR nanocomposites.

**Figure 13 polymers-11-01820-f013:**
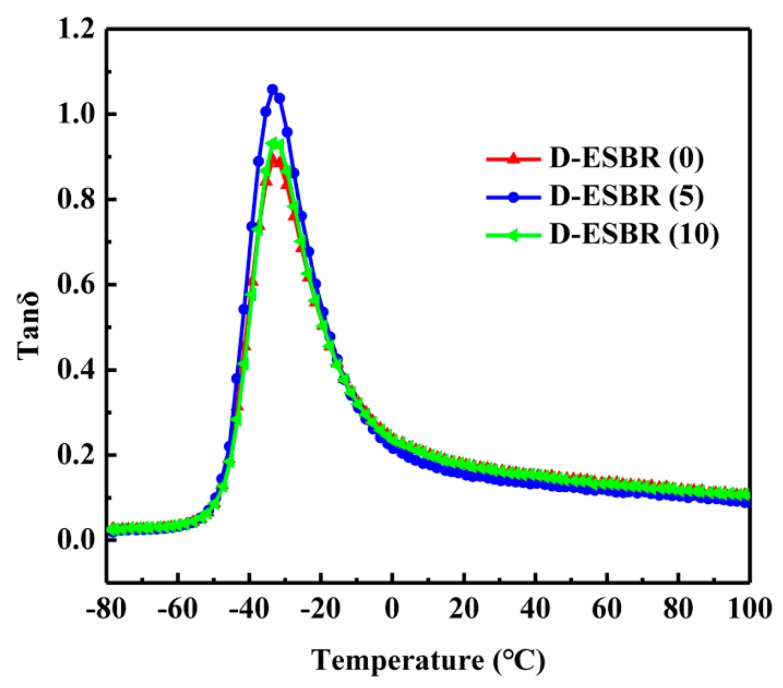
Temperature dependence of tan δ of silica/D-ESBR nanocomposites.

**Figure 14 polymers-11-01820-f014:**
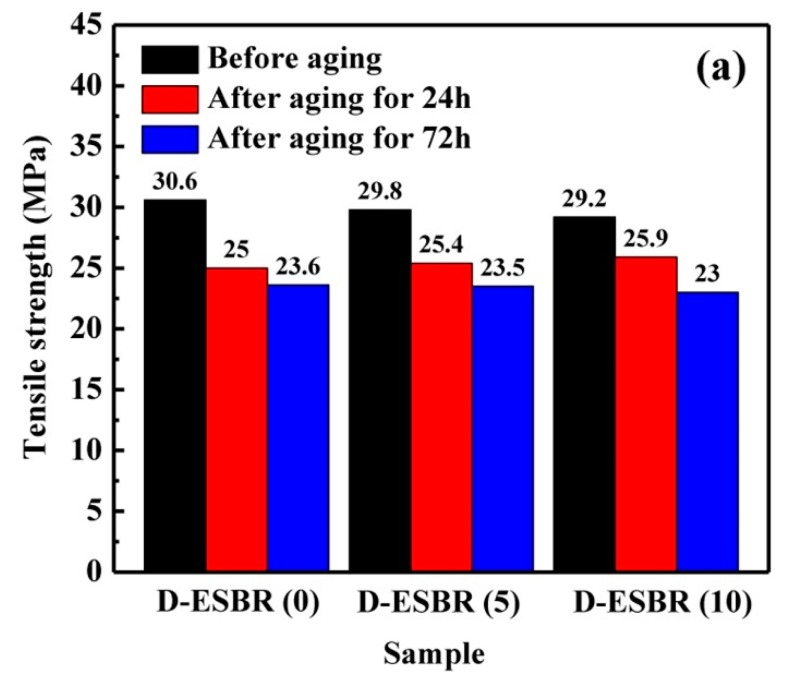
(**a**) Tensile strength and (**b**) elongation at break of D-ESBR nanocomposites before and after aging.

**Figure 15 polymers-11-01820-f015:**
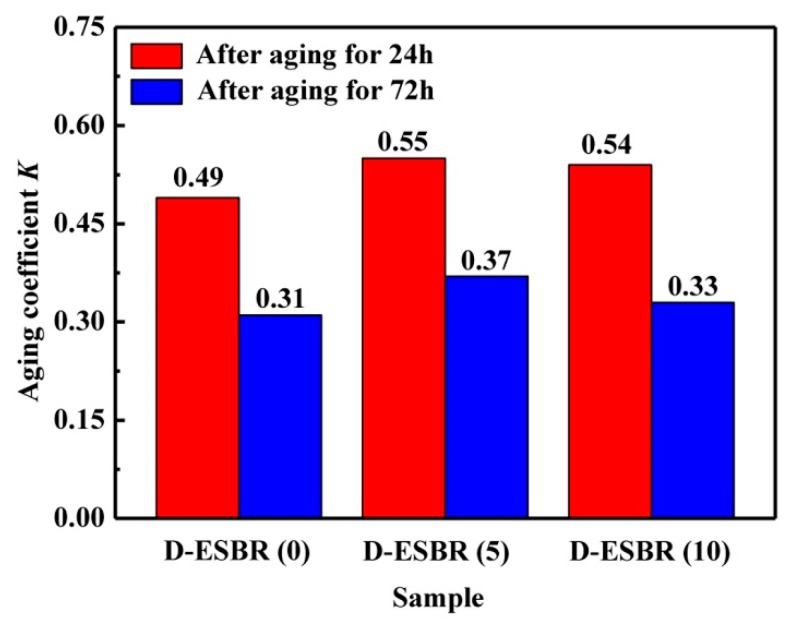
Aging coefficient *K* in D-ESBR nanocomposites at 24 and 72 h.

**Table 1 polymers-11-01820-t001:** Ingredients for the redox-initiated emulsion polymerization of D-ESBR.

Ingredients (Concentration)	Amount (g)
Dibutyl itaconate (DBI)	0/5/10
Styrene	30
Butadiene	70
Deionized water	150
Disproportionated potassium rosinate	5
Phosphoric acid	0.23
Potassium hydroxide	0.40
Ethylene diamine tetraacetic acid (EDTA)	0.03
Sodium dinaphthylmethanedisulfonate (TAMOL)	0.14
Sodium hydrosulfite (SHS)	0.04
Ferric ethylene diamine tetraacetic acid salt (Fe-EDTA)	0.02
Sodium hydroxymethanesulfinate (SFS)	0.04
Tert-dodecyl mercaptan (TDDM)	0.20
P-menthane hydroperoxide (PMH)	0.06

**Table 2 polymers-11-01820-t002:** Ingredients of Silica/D-ESBR nanocomposites.

Ingredients	Loading (phr) ^[a]^
D-ESBR ^[b]^	100
Silica (VN3)	60
Si69	6
Zinc oxide	5
Stearic acid	2
Antioxidant RD	1
Antioxidant 4020	1
Wax	1
Accelerator CZ	1.0
Accelerator NS	1.2
Sulfur	1.5

^[a]^ phr is the part per hundred of rubber; ^[b]^ D-ESBR with DBI contents of 0, 5, and 10 wt %. For instance, D-ESBR (5) stands for the D-ESBR obtained by adding 5 wt% of DBI to the polymerization mixture.

**Table 3 polymers-11-01820-t003:** Yields and molecular weights of D-ESBR.

Sample	DBI Content in Feed (wt%)	Yield (%)	Mn (g/mol)	PDI
D-ESBR (0)	0	70	196,647	3.11
D-ESBR (5)	5	72	191,202	2.56
D-ESBR (10)	10	75	187,614	2.54

**Table 4 polymers-11-01820-t004:** Percentages of itaconate, styrene, and butadiene moieties in D-ESBR.

Sample	Itaconate Moieties in D-ESBR (%)	Styrene Moieties in D-ESBR (%)	Different Configurations of Butadiene		
			*trans*- (%)	*cis*- (%)	*vinyl*- (%)
D-ESBR (0)	0	28.8	42.9	8.8	19.5
D-ESBR (5)	4.0	27.3	41.6	8.3	18.8
D-ESBR (10)	8.5	26.4	41.2	6.7	17.2

**Table 5 polymers-11-01820-t005:** Swelling ratio, cross-linking density of silica/D-ESBR samples.

Sample	Swelling Ratio (%)	Cross-Linking Density (10^−5^ mol/cm^3^)
D-ESBR (0)	159	15.3
D-ESBR (5)	198	13.6
D-ESBR (10)	254	9.7

**Table 6 polymers-11-01820-t006:** Tan δ values at a strain of 7% for the silica/D-ESBR nanocomposites.

Sample	Tan δ at a Strain of 7%
D-ESBR (0)	0.125
D-ESBR (5)	0.102
D-ESBR (10)	0.108

**Table 7 polymers-11-01820-t007:** Mechanical performance of the cross-linked silica/D-ESBR nanocomposites.

Properties	D-ESBR (0)	D-ESBR (5)	D-ESBR (10)
Tensile strength (MPa)	30.6 ± 0.4	29.8 ± 0.3	29.2 ± 0.1
Elongation at break (%)	586 ± 20	520 ± 31	628 ± 16
Modulus at 100%	2.2 ± 0.1	2.3 ± 0.1	2.0 ± 0.1
Modulus at 300%	11.8 ± 0.3	14.1 ± 0.6	10.8 ± 0.5
Permanent set (%)	10	8	12
Hardness (Shore A)	68.3	65.5	62.6
Akron abrasion loss	0.063	0.068	0.075
Heat build-up in compression	33.7	31.0	30.5

**Table 8 polymers-11-01820-t008:** Dynamic mechanical performance of the silica/D-ESBR nanocomposites.

Sample	*T*_g_ (°C)	Tan δ		
		0 °C	60 °C	Max
D-ESBR (0)	−33.2	0.239	0.135	0.89
D-ESBR (5)	−32.8	0.221	0.117	1.05
D-ESBR (10)	−32.1	0.238	0.132	0.93

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
