# Peer review of "Preparation and Performance of Silica/ESBR Nanocomposites Modified by Bio-Based Dibutyl Itaconate"

_polymers, 2019, doi:10.3390/polym11111820_

Round 1

Reviewer 1 Report

Although it is an interesting paper, it needs to be revised before further consideration for publication:

1- Authors should provide XRD results of samples with different concentrations as synthesized.

2- Authors should describe the application of their method in the industry. What are the potential uses of this nanocomposite. The tests provided are on characterization of samples but none of them is related to application. For instance if this product planned to be used outdoor, aging tests should be provided. The effects of environmental parameters are very important in such application.

3- Authors should improve the introduction section by discussing recent studies on synthesis of silica composites. Following studies can be included for such applications:

+ Polymer–silica composites and silicas produced by high-temperature degradation of organic component

Thermochimica Acta, Volume 615, 10 September 2015, Pages 43-50

+ Dispersibility of hydrophilic and hydrophobic nano-silica particles in polyethylene terephthalate films: evaluation of morphology and thermal properties

Polymers and Polymer Composites 23 (5), 285-296 (2015).

+ Various nano-silica particles affecting dyeability of poly (ethylene terephthalate)/silica nanocomposite films

Fibers and Polymers 14 (5), 743-751 (2013).

Author Response

Reviewer #1:

(1) Authors should provide XRD results of samples with different concentrations as synthesized.

Answer: Thanks so much for the comment. We have provided XRD results of D-ESBR with various DBI contents. Analysis has also been given in the revised manuscript.

For the nonlayered silica nanoparticle composites, XRD was commonly performed to analyze the degree of crystallinity of the nanocomposites. Figure 7 shows the XRD patterns of the silica/D-ESBR nanocomposites with various DBI contents. It can be seen that the broad diffraction peak at around 21° indicates that the D-ESBR is amorphous without crystalline structure, while the sharp peaks at 32o and 36o in the patterns relate to the additives such as zinc oxide.

Figure 7. XRD patterns of the silica/D-ESBR nanocomposites with various DBI contents.

(2) Authors should describe the application of their method in the industry. What are the potential uses of this nanocomposite. The tests provided are on characterization of samples but none of them is related to application. For instance if this product planned to be used outdoor, aging tests should be provided. The effects of environmental parameters are very important in such application.

Answer: Thanks so much for the comment. We have conducted thermo-oxidative aging experiments. Some statements have been added in the revised manuscript.

The effect of thermo-oxidative aging parameter is very important in the outdoor application, such as tire. Figure 14 presents the mechanical properties of D-ESBR nanocomposites before and after thermo-oxidative aging, and their aging coefficients at 100°C for 24h and 72h are shown in Figure 15. Silica/D-ESBR (5) and silica/D-ESBR (10) nanocomposites show higher aging coefficient than silica/D-ESBR (0) nanocomposite, indicating that the introduction of DBI can improve the anti-aging ability of silica/D-ESBR nanocomposites, because the introduction of DBI decreases the double bond content in D-ESBR macromolecules.

Figure 14. (a) Tensile strength and (b) elongation at break of D-ESBR nanocomposites before and after aging.

Figure 15. Aging coefficient K of D-ESBR nanocomposites for 24h and 72h.

(3) Authors should improve the introduction section by discussing recent studies on synthesis of silica composites. Following studies can be included for such applications:

+ Polymer-silica composites and silicas produced by high-temperature degradation of organic component

Thermochimica Acta, Volume 615, 10 September 2015, Pages 43-50

+ Dispersibility of hydrophilic and hydrophobic nano-silica particles in polyethylene terephthalate films: evaluation of morphology and thermal properties

Polymers and Polymer Composites 23 (5), 285-296 (2015).

+ Various nano-silica particles affecting dyeability of poly (ethylene terephthalate)/silica nanocomposite films

Fibers and Polymers 14 (5), 743-751 (2013).

Answer: Thanks so much for the comments. Related information has been added in the revised manuscript, see below:

Recently, the preparation of silica/polymer nanocomposites has received attention all over the world. For example, silica has been incorporated into polymers such as silica gels [8] and Poly(ethylene terephthalate) [9,10] to prepare silica/polymer nanocomposites, aiming to improve their overall performance.

[8] Krasucka, P.; Stefaniak, W.; Kierys, A.; Goworek, J. Polymer-silica composites and silica sproduced by high-temperature degradation of organic component. Thermochimica Acta 2015, 615, 43-50.

[9] Gashti, M.P.; Moradian, S.; Rashidi, A.; Yazdanshenas, M.E. Dispersibility of hydrophilic and hydrophobic nano-silica particles in polyethylene terephthalate films: evaluation of morphology and thermal properties. Polymers & Polymer Composites 2015, 23, 285-295.

[10] Gashti, M.P.; Moradian, S.; Rashidi, A.; Yazdanshenas, M.E. Various nano-silica particles affecting dyeability of Poly(ethylene terephthalate)/silica nanocomposite films. Fibers and Polymers 2013, 14, 743-751.

Reviewer #1:

(1) Authors should provide XRD results of samples with different concentrations as synthesized.

Answer: Thanks so much for the comment. We have provided XRD results of D-ESBR with various DBI contents. Analysis has also been given in the revised manuscript.

For the nonlayered silica nanoparticle composites, XRD was commonly performed to analyze the degree of crystallinity of the nanocomposites. Figure 7 shows the XRD patterns of the silica/D-ESBR nanocomposites with various DBI contents. It can be seen that the broad diffraction peak at around 21° indicates that the D-ESBR is amorphous without crystalline structure, while the sharp peaks at 32o and 36o in the patterns relate to the additives such as zinc oxide.

Figure 7. XRD patterns of the silica/D-ESBR nanocomposites with various DBI contents.

(2) Authors should describe the application of their method in the industry. What are the potential uses of this nanocomposite. The tests provided are on characterization of samples but none of them is related to application. For instance if this product planned to be used outdoor, aging tests should be provided. The effects of environmental parameters are very important in such application.

Answer: Thanks so much for the comment. We have conducted thermo-oxidative aging experiments. Some statements have been added in the revised manuscript.

The effect of thermo-oxidative aging parameter is very important in the outdoor application, such as tire. Figure 14 presents the mechanical properties of D-ESBR nanocomposites before and after thermo-oxidative aging, and their aging coefficients at 100°C for 24h and 72h are shown in Figure 15. Silica/D-ESBR (5) and silica/D-ESBR (10) nanocomposites show higher aging coefficient than silica/D-ESBR (0) nanocomposite, indicating that the introduction of DBI can improve the anti-aging ability of silica/D-ESBR nanocomposites, because the introduction of DBI decreases the double bond content in D-ESBR macromolecules.

Figure 14. (a) Tensile strength and (b) elongation at break of D-ESBR nanocomposites before and after aging.

Figure 15. Aging coefficient K of D-ESBR nanocomposites for 24h and 72h.

(3) Authors should improve the introduction section by discussing recent studies on synthesis of silica composites. Following studies can be included for such applications:

+ Polymer-silica composites and silicas produced by high-temperature degradation of organic component

Thermochimica Acta, Volume 615, 10 September 2015, Pages 43-50

+ Dispersibility of hydrophilic and hydrophobic nano-silica particles in polyethylene terephthalate films: evaluation of morphology and thermal properties

Polymers and Polymer Composites 23 (5), 285-296 (2015).

+ Various nano-silica particles affecting dyeability of poly (ethylene terephthalate)/silica nanocomposite films

Fibers and Polymers 14 (5), 743-751 (2013).

Answer: Thanks so much for the comments. Related information has been added in the revised manuscript, see below:

Recently, the preparation of silica/polymer nanocomposites has received attention all over the world. For example, silica has been incorporated into polymers such as silica gels [8] and Poly(ethylene terephthalate) [9,10] to prepare silica/polymer nanocomposites, aiming to improve their overall performance.

[8] Krasucka, P.; Stefaniak, W.; Kierys, A.; Goworek, J. Polymer-silica composites and silica sproduced by high-temperature degradation of organic component. Thermochimica Acta 2015, 615, 43-50.

[9] Gashti, M.P.; Moradian, S.; Rashidi, A.; Yazdanshenas, M.E. Dispersibility of hydrophilic and hydrophobic nano-silica particles in polyethylene terephthalate films: evaluation of morphology and thermal properties. Polymers & Polymer Composites 2015, 23, 285-295.

[10] Gashti, M.P.; Moradian, S.; Rashidi, A.; Yazdanshenas, M.E. Various nano-silica particles affecting dyeability of Poly(ethylene terephthalate)/silica nanocomposite films. Fibers and Polymers 2013, 14, 743-751.

Reviewer 2 Report

The authors reported the preparation and characterization of silica / poly(dibutyl itaconate-co-styrene-co-butadiene) nanocomposite (silica/D-ESBR) via redox emulsion polymerization in order to obtain “high performance-green” tire materials for automotive. In addition, they studied the physical and mechanical properties of silica/D-ESBR nanocomposites with different dibutyl itaconate contents.

I have just one recommendation for the authors:

At page 4, line 120: the authors should explain what is the used coupling agent Si69

Author Response

Reviewer#2:

At page 4, line 120: the authors should explain what is the used coupling agent Si69.

Answer: Thanks so much for the comment. We are sorry for the incomplete statement in the original manuscript. With the proposal of green tire conception, nonpetroleum-based silica has become one of the most widely used fillers in tire tread rubber. However, the surface of silica has a large number of hydroxyl groups, resulting in poor dispersion of silica in hydrophobic rubbers and poor compatibility with the rubber matrix. One of the efficient methods commonly used is the surface modification of silica with silane coupling agents. In this work, the silane coupling agent we used is bis-(g-triethoxysilylpropyl)-tetrasulfide (Si69), which is bifunctional silanes with both hydrolysable and organofunctional ends. Commonly, silica is in situ modified with the silane coupling agent through a reaction between the hydroxyl groups on the silica surface and the hydrolysable groups of the silane coupling agent during the compounding process, and then the modified silica is covalently bonded to the rubber matrix through a reaction between the organofunctional groups of the silane coupling agent and the carbon-carbon double bonds of the rubber during the vulcanization process. For giving readers a specific statement, we have revised the puzzling statement in the revised manuscript.

Page 4, Line 120: First, silica with a coupling agent (Si69) bis-(g-triethoxysilylpropyl)-tetrasulfide (Si69), which is bifunctional silanes with both hydrolysable and organofunctional ends, was kneaded with the D-ESBR in a Haake internal mixer for 8 min.

Reviewer 3 Report

The paper “Preparation and performance of robust silica/ESBR nanocomposites modified by bio-based dibutyl itaconate” describes the synthesis of a modified SBR rubber in emulsion. Nanosized silica is then added to the polymer, and a physico-mechanical characterization of the silica filled rubber is presented.

The addition of Itaconate is shown to reduce the crosslinking degree and to enhance the dispersion of silica nanoparticles. An optimum amount of dibutyl itaconate is found which optimize the dynamic-mechanical properties associated to tyre rolling resistance.

The paper is clearly written and deserve publication after some minor revisions based on the comments and detailed observations in the following

General comments

About the title: I do not understand why the adjective “robust” is adopted. The term is not reported any more in the paper, and it is not very clear to me the meaning of “robustness” in the context of the paper. In the paper (for example in the abstract, line 15) general reference to enhancement of ESBR is reported. In my opinion, it should be reported also the property or performance which is expected (or proved) to be enhanced by the ESBR modification and/or the addition of Silica.

Detailed observations

Line 30: consider substitution of “fuel economy” with “fuel efficient” Line 75: consider substitution of “Different” with “Differently” In Table 1, I’d suggest to indicate the different amount of Dybutil Itaconate instead of generally indicating “Various” Line 124: would it be possible to indicate which press was adopted for the curing at 15MPa? Line 171-172: consider rephrasing the sentence “The samples were prepared by fracturing the composites in liquid nitrogen and were previously sputter coated with gold.” In “The samples were prepared by fracturing the composites in liquid nitrogen. The fracture surfaces were sputter coated with gold before observation” Line 176-179. Which was the pretension applied to the samples during the DMA tests performed in tension mode? Line 188: consider substitution of “controlled” with “intentionally limited”. Further, the authors state that they limited the yield to avoid the formation of gels. Was a formation o gels observed in this work or was it reported already somewhere else? In the second case, would it be possible to add a reference? Line 222: the authors report a slight increase in Tg with increasing Itaconate content: in my opinion, the change in Tg is so limited to be included in the measurement precision. Unless the test were replicated on more than one sample (which is not reported in the paper) and the error in Tg cannot be shown to be less than 1°C, I’d conclude that the Itaconate content has no effect at all on Tg as measured by DSC. Line 243-245: “With the introduction of DBI, silica/D-ESBR (5) has a considerable crosslink density with silica/D-ESBR (0), which is silica/D-ESBR (10) slightly lower”. It seems to me that something is missing in this sentence. Please check it. Line 252-257: To my understanding, in this section the authors correlate the adsorption of accelerator by the ester group as the cause of a delay in crosslink reaction. Further, they state that this delay than brings to a lower crosslinking density. However, I don’t understand why the delay may be the cause of the reduced crosslinking: the curing reaction may be slowed down due to the reduction of accelerant, but it could get to the same crosslinking after a longer time. Line 263-265: In which sense a certain type of network (filler-filler o filler-rubber) is harmful or beneficial toward the DMA response? It should be made clear what “benefit” or “harm” mean in the context of the paper. Figure 8: i) please modify the y-axis caption of figure 8a from “Tanδ” to “tanδ”. ii) Why the x-axis scale is different in figure 8a and 8b? I’d expect the scale to be the same. iii) The trend reported in this figure are well commented in the text. However slope of G’ for D-ESBR(5) at low strain is significantly different than that of the other compounds. This feature is not commented: is there any explanation/interpretation to this feature? Line 285: Authors report the value of tanδ a 7% strain to be significantly low in case on D-ESBR(5), and to be promising with respect of a reduced rolling resistance. However, the importance of this point cannot be fully understood by readers that are not familiar with the specific topic. In order to enhance general public understanding, it might be useful to report as a benchmark some figures from other materials that are already adopted in green tyres. A similar comment applies to the tanδ data from temperature ramps at the 7% strain amplitude. The values of tanδ at 60°C could be compared with data from literature to enable the general public to evaluate the degree of enhancement of the materials investigated.

Author Response

Reviewer#3:

General comments

(1) About the title: I do not understand why the adjective “robust” is adopted. The term is not reported any more in the paper, and it is not very clear to me the meaning of “robustness” in the context of the paper.

Answer: Thanks so much for the comment. We are sorry for the puzzling statement. We have corrected the puzzling statement based on the comment.

Page 1, Line 1: Preparation and performance of robust silica/ESBR nanocomposites modified by bio-based dibutyl itaconate.

(2) In the paper (for example in the abstract, line 15) general reference to enhancement of ESBR is reported. In my opinion, it should be reported also the property or performance which is expected (or proved) to be enhanced by the ESBR modification and/or the addition of Silica.

Answer: Thanks so much for the comments. Related information has been added in the revised manuscript, see below:

Page 1, Line 15: Nonpetroleum-based silica with hydroxy groups was used as a filler to enhance the D-ESBR, which can provide excellent mechanical properties, low rolling resistance, and high wet skid resistance.

Detailed observations

(3) Fix up some expressions.

Line 30: consider substitution of “fuel economy” with “fuel efficient”. Line 75: consider substitution of “Different” with “Differently”. In Table 1, I’d suggest to indicate the different amount of Dybutil Itaconate instead of generally indicating “Various”. Line 171-172: consider rephrasing the sentence “The samples were prepared by fracturing the composites in liquid nitrogen and were previously sputter coated with gold.” In “The samples were prepared by fracturing the composites in liquid nitrogen. The fracture surfaces were sputter coated with gold before observation”. Line 188: consider substitution of “controlled” with “intentionally limited”.

Answer: Thanks so much for the comment. We are sorry for the inappropriate statements we made in the manuscript. We have corrected the inappropriate statements based on the comment.

(a) Page 1, Line 30: With the international calls for energy conservation, environmental protection, and emission reduction, a growing voice to make motor vehicles more fuel economy fuel efficient is widespread all over the world, thus calling for a high-performance tire, i.e. green tire [1].

(b) Page 2, Line 75: Different Differently from the previous introduced petroleum-based functional monomers, dibutyl itaconate has two ester groups located on different sides, which means dibutyl itaconate can introduce higher polar group density than those petroleum-based monomers under the same conditions.

(c) Page 4, Line 116: The different amount of Dybutil Itaconate have been added in Table 1 instead of“Various”.

Table 1. Recipe for redox-initiated emulsion polymerization of D-ESBR.

Ingredients (concentration)

Amount (g)

Dibutyl itaconate

Various 0/5/10

Styrene

30

Butadiene

70

Deionized water

150

Disproportionated potassium rosinate

5

Phosphoric acid

0.23

Potassium hydroxide

0.40

EDTA

0.03

TAMOL

0.14

SHS

0.04

Fe-EDTA

0.02

SFS

0.04

TDDM

0.20

PMH

0.06

(d) Page 6, Line 171-172: The samples were prepared by fracturing the composites in liquid nitrogen and were previously sputter coated with gold. The samples were prepared by fracturing the composites in liquid nitrogen. The fracture surfaces were sputter coated with gold before observation.

(e) Page 6, Line 188: The yield of D-ESBR was controlled intentionally limited in 70-80% to obtain optimum mechanical properties because the D-ESBR with higher yield than 80% has a high gel content.

(4) Line 124: would it be possible to indicate which press was adopted for the curing at 15MPa?

Answer: Thanks so much for the comment. Related information has been added in the revised manuscript, see below:

Page 4, Line 116: Finally, the silica/D-ESBR compound was cured in a XLB-D 350 × 350 hot press (Huzhou Eastmachinery Corporation, China) under 15 MPa at 150℃ to prepare silica/D-ESBR nanocomposite.

(5) Line 176-179. Which was the pretension applied to the samples during the DMA tests performed in tension mode?

Answer: Thanks so much for the comment. In tension mode, we adjusted the pretension by controlling "displacement", which the value is generally between -100 and -50, and the displacement is adjusted at -70 during the test. This displacement is represented the pretension as mentioned.

(6) Line 188: The authors state that they limited the yield to avoid the formation of gels. Was a formation of gels observed in this work or was it reported already somewhere else? In the second case, would it be possible to add a reference?

Answer: Thanks so much for the comment. Actually, we can observe gels obviously when the yield reach up to 80%. The related statement has been added into the revised manuscript.

(7) Line 222: the authors report a slight increase in Tg with increasing Itaconate content: in my opinion, the change in Tg is so limited to be included in the measurement precision. Unless the test were replicated on more than one sample (which is not reported in the paper) and the error in Tg cannot be shown to be less than 1°C, I’d conclude that the Itaconate content has no effect at all on Tg as measured by DSC.

Answer: Thanks so much for the comment. We are sorry for the unclear statement that the introduction of DBI slightly increases the Tg of the D-ESBR. We have corrected the unclear statement based on the comment.

Page 8, Line 222: The introduction of DBI slightly increases the Tg of the D-ESBR. The introduction of DBI nearly has little effect on Tg of the D-ESBR.

(8) Line 243-245: “With the introduction of DBI, silica/D-ESBR (5) has a considerable crosslink density with silica/D-ESBR (0), which is silica/D-ESBR (10) slightly lower”. It seems to me that something is missing in this sentence. Please check it.

Answer: Thanks so much for the comment. We are sorry for the puzzling statement in the original manuscript.

With the introduction of DBI, silica/D-ESBR (5) has a considerable crosslink density with silica/D-ESBR (0), which is silica/D-ESBR (10) slightly lower.

With the introduction of DBI, the torque difference gradually decreases, indicating the decrease of cross-linking density of silica/D-ESBR.

(9) Line 252-257: To my understanding, in this section the authors correlate the adsorption of accelerator by the ester group as the cause of a delay in crosslink reaction. Further, they state that this delay than brings to a lower crosslinking density. However, I don’t understand why the delay may be the cause of the reduced crosslinking: the curing reaction may be slowed down due to the reduction of accelerant, but it could get to the same crosslinking after a longer time.

Answer: Thanks so much for the comment.

ester groups may absorb the accelerators, which decreases the cross-linking density by retarding the curing process.

With the introduction of DBI, ester groups of DBI may absorb a portion of the accelerators resulting in forming more polysulfide bond and disulfide bond during the curing system. The total of polysulfide bond density and disulfide bond density increases will lead to a decrease in cross-linking density.

(10) Line 263-265: In which sense a certain type of network (filler-filler or filler-rubber) is harmful or beneficial toward the DMA response? It should be made clear what “benefit” or “harm” mean in the context of the paper.

Answer: Thanks so much for the comment. Filler-filler network is harmful to the DMA response, which means filler

We are sorry for the puzzled statement. The words “benefit” and “harm” are not accurate expression toward dynamic response. Acturally, we aimed to investigate the silica dispersion and interfacial interaction of the nanocomposites, in which the filler-filler and filler-rubber network can be used to evaluate the silica dispersion and interfacial interaction of the nanocomposites. Also, the related statement below has been added into the revised manuscript.

Generally, the formation of the filler-filler network results in poor silica dispersion and interfacial interaction, while the formation of the filler-rubber network results in improved silica dispersion and interfacial interaction.

(11) Figure 8:

i) please modify the y-axis caption of figure 8b from “Tanδ” to “tanδ”. ii) Why the x-axis scale is different in figure 8a and 8b? I’d expect the scale to be the same.

iii) The trend reported in this figure are well commented in the text. However slope of G’ for D-ESBR(5) at low strain is significantly different than that of the other compounds. This feature is not commented: is there any explanation/interpretation to this feature?

Answer: Thanks so much for the comment.

i) The figure 8b has been revised according to the comment. ii) For the rubber compounds, the strain varied from 0.1%-400% at the test frequency of 1 Hz. For the rubber vulcanizates, the strain varied from 0.1%-42% at the test frequency of 1 Hz. Therefore, the x-axis scale is different in figure 8a and 8b, and we have been modified the x-axis scale above Figure.

iii) Slope of G’ for silica/D-ESBR (5) at low strain is significantly different than that of the other compounds, Given explanations as following: owning to the hydrogen bonding, filler-filler network of the silica/D-ESBR (5) is weaken, so the Payne effect is significantly weakened at low strain. Compared with the silica/D-ESBR (5), however, the silica/D-ESBR (10) has higher hydrogen bond density, which is also contributed to the G’ value. The weak hydrogen bonds are easily destroyed under low strain shear, resulting in more decrease in G’ value than that of silica/D-ESBR (5).

(12) Line 285: Authors report the value of tanδ at 7% strain to be significantly low in case on D-ESBR(5), and to be promising with respect of a reduced rolling resistance. However, the importance of this point cannot be fully understood by readers that are not familiar with the specific topic. In order to enhance general public understanding, it might be useful to report as a benchmark some figures from other materials that are already adopted in green tyres. A similar comment applies to the tanδ data from temperature ramps at the 7% strain amplitude. The values of tanδ at 60°C could be compared with data from literature to enable the general public to evaluate the degree of enhancement of the materials investigated.

Answer: Thanks so much for the comment.

The tanδ value at 7% strain of the silica/G-ESBR nanocomposites is between 0.120 and 0.331 [27].

The tanδ values at 60°C of the silica/G-ESBR nanocomposites is between 0.108 and 0.123 [31].

27. Qiao, H.; Chao, M.; Hui, D.; Liu, J.; Zheng, J.; Lei, W.; Zhou, X.; Wang, R.; Zhang, L. Enhanced interfacial interaction and excellent performance of silica/epoxy group-functionalized styrene-butadiene rubber (SBR) nanocomposites without any coupling agent. Compos. Part B Eng. 2017, 114, 356-364.

31. Mun, H.; Hwang, K.; Yu, E.; Kim, W.; Kim, W. Glycidyl methacrylate-emulsion styrene butadiene rubber (GMA-ESBR)/silica wet masterbatch compound. Polymers. 2019, 11, 1000.

Round 2

Reviewer 1 Report

Based on my previous comments and the revised version, authors considered new experimental results and the paper is acceptable now.